# Tandem Transesterification–Esterification Reactions Using a Hydrophilic Sulfonated Silica Catalyst for the Synthesis of Wintergreen Oil from Acetylsalicylic Acid Promoted by Microwave Irradiation

**DOI:** 10.3390/molecules27154767

**Published:** 2022-07-26

**Authors:** Sandro L. Barbosa, David Lee Nelson, Milton de S. Freitas, Wallans Torres Pio dos Santos, Stanlei I. Klein, Giuliano C. Clososki, Franco J. Caires, Alexandre P. Wentz

**Affiliations:** 1Department of Pharmacy, Universidade Federal dos Vales do Jequitinhonha e Mucuri-UFVJM, Campus JK, Rodovia MGT 367—Km 583, n^o^ 5.000, Alto da Jacuba, Diamantina 39100-000, Brazil; dleenelson@gmail.com (D.L.N.); freitas.milton@hotmail.com (M.d.S.F.); wallanst@ufvjm.edu.br (W.T.P.d.S.); 2Department of General and Inorganic Chemistry, Institute of Chemistry, São Paulo State University-Unesp, R. Prof. Francisco Degni 55, Quitandinha, Araraquara 14800-900, Brazil; stanleiklein@gmail.com; 3Department of Physics and Chemistry, Faculdade de Ciências Farmacêuticas de Ribeirão Preto, São Paulo University-USP, Av. do Café s/n, Ribeirão Preto 14040-903, Brazil; gclososki@usp.br (G.C.C.); fjcaires@usp.br (F.J.C.); 4Centro Universitário SENAI-CIMATEC, Av. Orlando Gomes, 1845, Piatã, Salvador 41650-010, Brazil; alexandre.wentz@fieb.org.br

**Keywords:** sulfonated silica catalyst (SiO_2_–SO_3_H), methyl salicylate, green chemistry, methylating agent, deacetylation (acyl nucleophilic substitution), solid acid catalyst

## Abstract

SiO_2_–SO_3_H, with a surface area of 115 m^2^/g and pore volume of 0.38 cm^3^g^−1^, and 1.32 mmol H^+^/g was used as a 20% *w*/*w* catalyst for the preparation of methyl salicylate (wintergreen oil or MS) from acetylsalicylic acid (ASA). A 94% conversion was achieved in a microwave reactor over 40 min at 120 °C in MeOH. The resulting crude product was purified by flash chromatography. The catalyst could be reused three times.

## 1. Introduction

Wintergreen (*Gaultheria procumbens* L.), also known as checkerberry or teaberry, is a small, ericaceous plant found growing in the undergrowth of dense forests in the U.S. Wintergreen is cultivated for use in the landscape industry, and it is the source of the essential oil of wintergreen (WO) [1,2]. The essential oil of wintergreen is prepared commercially by steam distillation; however, the most commonly used form of WO is synthetic. Wintergreen is now commonly used as a flavoring agent, but its leaves were historically used by North American natives for the treatment of aches and pains due to their “aspirin-like” quality. In fact, WO, the most common salicylate in commercial wintergreen preparations, and is routinely used in topical ointments for the treatment of inflammation [2].

WO is a clear liquid with a peppermint and minty scent, which most likely serves as a defense against herbivores. For example, when a plant is infected with herbivorous insects, the production of WO attracts other insects that kill these herbivorous insects [3]. WO can also be used by plants as a pheromone to warn other plants of pathogens, such as the tobacco mosaic virus [4], and it is used as a clearing agent for preparing slides of *Aedes* mosquito larvae for microscopic examination [5]. It is also extensively used in the synthesis of solvents, perfumes, cosmetics, food preservatives, chiral auxiliaries, plasticizers, drugs, and pharmaceuticals [6,7]. Ribnicky et al., (2003) determined the presence of salicylates, other than methyl salicylate, that could act as alternatives for aspirin [8]. Recently, the authenticity of methyl salicylate (MS) in the essential oils from *Gaultheria procumbens* L. and *Betula lenta* L. was determined using isotope ratio mass spectrometry [9].

Among the methods found in the literature for the synthesis from salicylic acid (SA), the preparation of methyl salicylate (MS) using diazotization chemistry is emphasized [10]. In this previous experiment, the use of diazonium salts for the replacement of an aromatic amine group by a phenolic hydroxyl was demonstrated. Many reports have been published on the synthesis of WO by esterification of SA with dimethyl carbonate (DMC), such as, “Green synthesis of WO using novel sulfated iron oxide-zirconia catalyst” [11]. Sreekumar et al., reported the reaction of SA with DMC using zeolite, wherein monomethylation was observed [12]. Kirumakki et al., obtained a 90% conversion of salicylic acid and 95% selectivity in the reaction of SA with DMC over zeolites when treated for 4 h at 135 °C [13]. Zheng et al., reported 98% conversion and 96% selectivity with DMC over the AlSBA15-SO_3_H catalyst when treated for 8 h at 200 °C [14]. Su et al., obtained 99% conversion of SA and 77% selectivity for the reaction of SA with DMC over mesoporous aluminosilicate [15], and Zhang et al., reported 93% conversion and 99% selectivity with methanol using Ce^4+^-modified cation exchange resins when treated for 12 h at 95 °C [16]. In all of these studies, equilibrium in the esterification reaction was overcome using dimethyl carbonate, instead of methanol, as the methylation agent.

Some studies have reported the formation of MS from the esterification reaction of SA with MeOH as a methylation agent, using different solid catalysts. Hua Shi et al., reported that a variety of Brønsted acidic ionic liquids were screened as catalysts for the esterification of salicylic acid in a microwave-accelerated process [17].

Esterification or transesterification using microwave irradiation, in addition to being environmentally friendly, is also marked by a considerable reduction in reaction time in comparison with conventional esterification [18,19]. Furthermore, to the best of our knowledge, tandem esterification-transesterification of acetylsalicylic acid (ASA) in hydrophilic sulfonated silica catalyst has not yet been achieved. There are few studies that have reported the transesterification/esterification of ASA. No report of the use of liquid or solid catalysts using the principles of Green Chemistry has been published.

As a part of an ongoing research on the use of the SiO_2_–SO_3_H catalyst for clean synthesis [20] using the principles of Green Chemistry, this catalyst was used in the highly selective, one-pot, tandem, transesterification–esterification reactions of ASA with MeOH in the microwave-accelerated synthesis of MS. A high catalytic activity in a very short period of time in an inexpensive process to produce high yields of highly pure MS from ASA was observed. The synthesis of MS directly by methylation of salicylic acid was also achieved. The stability and re-use of SiO_2_–SO_3_H was also examined.

## 2. Experimental

### 2.1. Raw Materials and Chemicals

All the reagents (analytical grade), including ASA, were supplied by Vetec, São Paulo, Brazil, and were used without further purification.

### 2.2. Instrumentation

All the reactions were monitored by GC–MS. The compositions of the reaction products were determined on a GC–MS-QP 2010/AOC 5000 AUTO INJECTOR/Shimadzu Gas Chromatograph–Mass Spectrometer equipped with a 30 m Agilent J&W GC DB-5 MS column (Santa Clara, CA, USA). Direct insertion spectra were measured at 70 eV. Quantitative analyses were performed on a Shimadzu GC-2010 gas chromatograph (Kyoto, Japan) equipped with a flame ionization detector under the same conditions as specified for the GC–MS analyses [20]. We also used a GC–FID, equipped with a 30 m Agilent J&W GC DB-17 MS column. Final hold time was approximately 20 min, where the temperature ramp started at 70 °C for 5 min, reaching 270 °C, remaining at this temperature for 5 min using nitrogen as the carrier gas (See Appendix A). ^1^H- and ^13^C-NMR spectra were recorded on Bruker Avance 400 Spectrometers (Billerica, MA, USA) using deuterated methanol as the solvent and TMS as the internal reference [20]. The purification of the products was achieved by flash column chromatography using a mixture of hexane/ethyl acetate in a 9/1 proportion as the eluent. The MW reactions were performed in 10 mL G-10 vials of an Anton Paar single-mode MW Monowave 300 synthesis reactor (Graz, Austria), powered by an 850 W magnetron, and equipped with temperature sensor and magnetic stirring [21].

### 2.3. Preparation of the Silica Gel and Sulfonated Silica (SiO_2_–SO_3_H)

The preparation of silica gel and the sulfonated silica, SiO_2_–SO_3_H, catalyst have been reported previously [20]. Three hundred grams of sand and 600.0 g of sodium carbonate were homogenized and transferred to porcelain crucibles, which were heated at 850 °C for 4 h. The hot solid mixtures were transferred to a glass fritted filter and washed with 600–900 mL of boiling water. The filtered solution was acidified to pH 1 with hydrochloric acid, and the white precipitate was filtered and dried at 400 °C. The resulting silica was passed through a 24-mesh sieve for standardization. Ten grams of the prepared silica was mixed with 10.0 mL of H_2_SO_4_, and the mixture was stirred at room temperature for 12 h, filtered, dried at 150 °C for 4 h, cooled, and stored in a desiccator. The acid strength of 1.32 mmol of H^+^ per gram of catalyst was determined by potentiometric titration. The surface area was determined to be 115 m^2^/g, and the pore volume was measured as 0.38 cm^3^g^−1^.

### 2.4. Typical Procedures

#### 2.4.1. Tandem Esterification and Transesterification of ASA in MeOH Using SiO_2_–SO_3_H as the Catalyst

To a 10 mL microwave reactor vial, 180.0 mg (1.00 mmol) of ASA (purified by column chromatography), 1.0 mL of MeOH, and 0.0360 g of SiO_2_–SO_3_H (20% *w*/*w* in relation to ASA) was added. The vial was heated in the microwave reactor at 120 °C for 40 min. The mixture was cooled to room temperature, and 10.0 mL of CH_2_Cl_2_ was added. The organic solution obtained after filtration of the solid catalyst was transferred to an extraction funnel and partitioned between 10 mL of CH_2_Cl_2_ and 20 mL of saturated NaHCO_3_, dried with magnesium sulfate, filtered, and evaporated under reduced pressure. The resulting residue was subjected to GC–MS analysis, which demonstrated the absence of unreacted SA. The residue was then purified by flash column chromatography on silica using hexane–ethyl acetate (9:1) as the mobile phase to yield MS as a colorless oil. Three more runs could be performed with the same catalyst. Each run was performed in triplicate.

#### 2.4.2. Esterification (Direct Methylation) of SA in MeOH Using SiO_2_–SO_3_H as the Catalyst

To a 10 mL microwave reactor vial, 138.1 mg (1.00 mmol) of SA, 1.0 mL of MeOH, and 27.6 mg of SiO_2_–SO_3_H (20% *w*/*w* in relation to SA) were added. The vial was heated in the microwave reactor at 120 °C for 40 min. The mixture was cooled to room temperature, and the product was isolated as described above (Section 2.4.1) The resulting residue was subjected to GC–MS analysis, which demonstrated the presence of 1.4% of unreacted SA and 98.6% of MS. The residue was then purified by flash column chromatography on silica using hexane–ethyl acetate (9:1) as the mobile phase to yield MS as a colorless oil. 

## 3. Results and Discussion

One of the first syntheses of MS from ASA (contained in commercial aspirin tablets) was reported by Aaron M. Hartel and James M. Hanna Jr. as a short, single-pot preparation that can be performed via a tandem transesterification–Fischer esterification. The crushed aspirin tablets were mixed with MeOH to dissolve the aspirin, and insoluble material was removed by filtering through a cotton plug. Concentrated H_2_SO_4_ was added to the aspirin solution, and it was either refluxed for 90 min or heated in a scientific microwave system at 120 °C for 5 min. The MS was obtained in yields approaching 70% [22].

Esterification reactions are reported with various homogeneous acid catalysts, such as HCl, H_2_SO_4_, HF, and H_3_PO_4_. However, these methods have drawbacks, such as the generation of undesired inorganic salts, hazardous conditions, difficulty in catalyst recovery, and limited reusability [23,24].

The SiO_2_–SO_3_H catalyst has been used in our research as a replacement for conventional mineral acid catalysts (H_2_SO_4_), in various studies involving the esterification of different carboxylic acids [20]. However, the use of this catalyst in transesterification reactions has only been tested with triglycerides [25]. In the present case, the reaction proceeds via a tandem transesterification–esterification.

In this work, we studied the rapid synthesis of MS from ASA in a single reaction vessel, where the SiO_2_–SO_3_H catalyzed the acyl substitution on the carboxylic group of ASA with MeOH as the nucleophilic reagent (the methylating agent) to produce an ester. Simultaneously, the transesterification of the acetyl group with the nucleophilic MeOH yielded methyl acetate as a byproduct and MS (principal product, Figure 1).

The ASA easily absorbs humidity, and intramolecular catalysis leads to the formation of acetic acid and salicylic acid (Equation (1)) [24]. Therefore, the ASA was purified by recrystallization using aqueous ethanol prior to use. The NMR spectrum demonstrating the purity of the ASA used for the synthesis, is shown in Figure 2 [26].
ASA + H_2_O → SA + CH_3_COOH(1)

The tandem transesterification–esterification reaction was tested under different conditions using a microwave reactor as a heating source. Triplicate results (entries 8, 8′, and 8″) were obtained to evaluate reproducibility using the best reaction condition obtained (Table 1).

In a reaction medium containing SiO_2_–SO_3_H, the acetate ester group of ASA is initially transesterified to form methyl acetate and SA (methanolysis). Under the same conditions, the carboxylic acid group of the resulting salicylic acid undergoes Fischer type esterification to form MS. The procedure produces MS at a high yield (94%) after 40 min in a microwave-irradiated process (850 W), at atmospheric pressure and 120 °C, of high purity, as determined by ^1^H NMR (Figure 3) and ^13^C NMR (Figure 4).

The tandem transesterification–esterification reaction was confirmed by monitoring the reaction process by gas chromatography from the first minutes of the reaction. We detected the total conversion of ASA into SA (89.10% yield). The attack of MeOH on the acetyl group to yield free SA (Equation (2)) is normally catalyzed intramolecularly by the ortho-carboxylate group [27]. However, under the conditions utilized in this study, 25.75% conversion of ASA to SA in the absence of the solid catalyst was observed, but no MS was formed with microwave irradiation during one minute at 120 °C. The transesterification reaction is totally favored by the catalysis by SiO_2_–SO_3_H, and it is complete within one minute.
ASA + MeOH → SA + AcOMe(2)

The esterification reaction (Equation (3)) catalyzed by the SiO_2_–SO_3_H also initiates within one minute, and MS (10.90% yield) is formed. After five minutes of reaction, the reaction medium consisted of SA (80.00% yield) and MS (20.00% yield). At the end of 40 min, the reaction medium contained SA (6.33% yield) and MS (93.67% yield). At this point, the reaction was terminated.
SA + MeOH → MS + H_2_O(3)

Kirumakki et al. [13] reported that the conventional esterification reaction using acid and alcohol results in the formation of water as a co-product, and leads to leaching of active sites in many liquid-phase reactions. This leaching process decreases the activity of the catalyst. As can be seen from the yield (94%) obtained in the MS synthesis from ASA, these drawbacks were minimized when SiO_2_–SO_3_H was used as the catalyst. The catalyst was reused three times to provide 90%, 85%, and 78% yields of MS.

We performed various reactions using these conditions, including the esterification of SA in MeOH using SiO_2_–SO_3_H as the catalyst. A 98.6% yield was obtained in the esterification reaction. 

Greenness and efficiency are two principal issues in this tandem process. These issues can be characterized in terms of atom economy, as determined in Green Chemistry Principle 2 [28]. In this work, the two reactions with no byproducts are desirable, and clean and reliable reactions must be employed when planning the synthesis of MS. Step economy is another fundamental aspect to consider when minimizing the number of reaction steps to MS. Reducing the number of steps reduces the length, cost, development time, execution time, effort, number of separation methods, and environmental impact of MS synthesis. Step economy was clearly influenced by selecting the right reaction method and sequence to allow for an optimal increase in target-relevant complexity. In this process, two reactions were performed in the same microwave reactor over a 40-min period. The occurrence of two reactions in a single reactor, without isolating or purifying the intermediates (SA), is important for greenness and practicability in synthesizing MS.

## 4. Conclusions

The use of microwave irradiation decreased the time required for the production of MS by transesterification–esterification of ASA with MeOH at 120 °C by a factor of four in the presence of the SiO_2_–SO_3_H catalyst, and a 94% conversion and 100% selectivity were observed. The reuse of the SiO_2_–SO_3_H catalyst, and the immediate use of the intermediate formed in the process, which involves one less step, represent the economy of the process, and a conformity to the principles of Green Chemistry. The tandem reaction led to the study of the use of the catalyst for the direct methylation of SA with MeOH (98.6% yield).

## Figures and Tables

**Figure 1 molecules-27-04767-f001:**
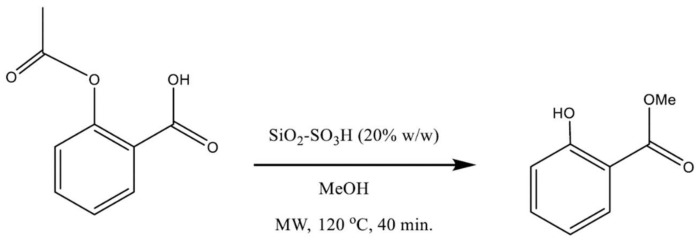
Reactions of the conversion of ASA into MS. The synthesis proceeds via the tandem transesterification–esterification reaction catalyzed by SiO_2_–SO_3_H.

**Figure 2 molecules-27-04767-f002:**
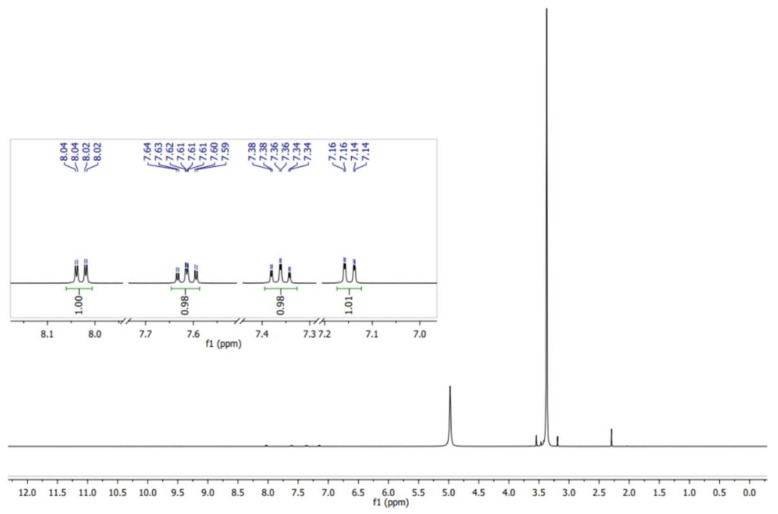
^1^H NMR spectrum (in MeOH) of the purified ASA utilized for the study of the transesterification–esterification reaction catalyzed by SiO_2_–SO_3_H.

**Figure 3 molecules-27-04767-f003:**
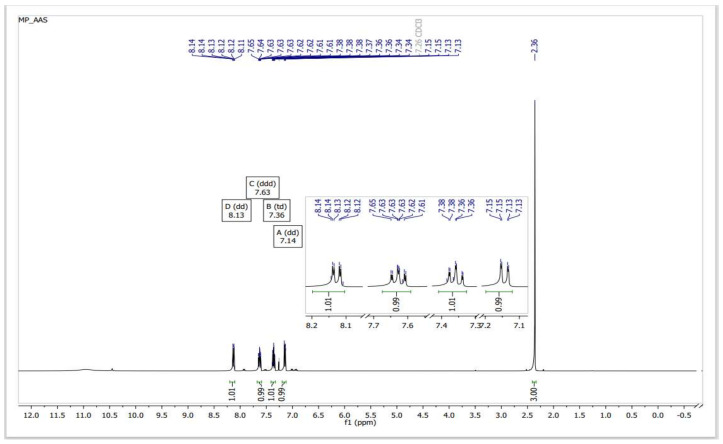
^1^H NMR spectra of a sample of MS obtained from the transesterification–esterification of ASA catalyzed by SiO_2_–SO_3_H.

**Figure 4 molecules-27-04767-f004:**
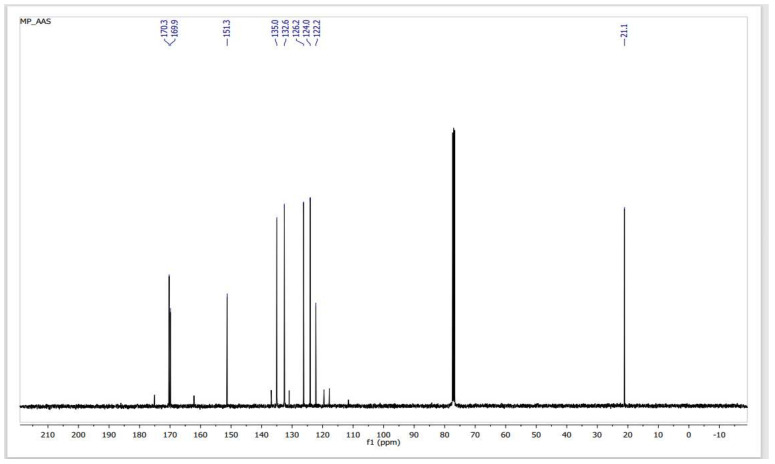
^13^C NMR spectra of a sample of MS obtained from the transesterification–esterification of ASA catalyzed by SiO_2_–SO_3_H

**Table 1 molecules-27-04767-t001:** The experimental results obtained for the transesterification–esterification of ASA.

Entry ^a^	MeOH (mL)	SiO_2_–SO_3_H Catalyst (%)	Time (min)	Temperature (°C)	Yield (%)
1	1.0	10	10	120	60
2	1.0	10	20	120	56
3	1.0	20	5	120	23
4	0.5	20	10	120	52
5	1.0	20	10	120	62
6	1.0	20	20	120	54
7	2.0	20	20	120	37
8 ^b^	1.0	20	40	120	94
8′ ^b^	1.0	20	40	120	93
8″ ^b^	1.0	20	40	120	93

a—MW: Anton Paar Monowave 300 microwave reaction, non-inert conditions, 850 W power. b—8, 8′ and 8″ refer to three different reactions under the same conditions.

## Data Availability

Data is available from the authors upon request.

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
