# Peer review of "Tandem Transesterification–Esterification Reactions Using a Hydrophilic Sulfonated Silica Catalyst for the Synthesis of Wintergreen Oil from Acetylsalicylic Acid Promoted by Microwave Irradiation"

_molecules, 2022, doi:10.3390/molecules27154767_

Round 1

Reviewer 1 Report

The authors investigated the tandem transesterification/esterification of acetylsalicylic acid with methanol under heterogeneous catalysis using SiO2-SO3H.  The work is carried out well and produces the intended product (methyl salicylate) in good yield and purity.  I would suggest the following be taken into account to improve the manuscript:

1.  The authors do not report the preparation of the catalyst, but instead provide a reference to previous work.  Given that the catalyst is critical to the current work, I would suggest this be included in the Experimental.

2.  The abstract provides details about the catalyst (surface area, activity, etc.) that is not reported in the manuscript.  This should be included in the Experimental as suggested in point 1, above.  The activity, 1.32 mmol H+/g is especially relevant and important.

3.  The abstract suggests that the catalyst can be re-used 3 times.  This is not discussed in the manuscript but is an important consideration relative to green chemistry.  The authors do say that triplicate results were obtained (Table 1, entry 8, 8' and 8''), but it is not clear whether the same catalyst or fresh catalyst was used for these three runs.  In a related issue, the end of the introduction says that the stability and re-use of the catalyst was examined, but this is not discussed in the manuscript.

4.  the abbreviation for acetylsalicylic acid (ASA) is not properly defined in the introduction.  The term ASA is used without defining it.

5.  Why not demonstrate that MS can be directly methylated under these conditions since ASA is actually derived from MS  which is formed by the Kolbe reaction on an industrial scale.  The direct methylation is at least equally important, if not more so, than forming MS from aspirin.

6.  Are there no other examples that can be shown to take place via this same tandem reaction?  This would strengthen the significance of the manuscript.

Reviewer 2 Report

Manuscript presented by Sandro L. Barbosa et al. presented presented a synthesis method of methyl salicylate from acetylsalicylic acid promoted by microwave irradiation. The manuscript is written and prepared in neutral way, not too bad. Easy to read and follow. Some aspects should be improved (or need to be re-written).

I recommend the article to publish but first the paper should be improve. My decision – reconsider after major revision. Comments to be considered, in order to further improve the manuscript quality:

(1)   Would you explicitly specify the novelty of your work? In the introduction it is not clearly presented. Please comment it in the introduction.

(2)   Scheme “SA + MeOH → MS” and all should be separate, not write in text.

(3)   Please complete the hardware specification in “Instrumentation” part, eg. no have information about MS ionization detector, internal standard of NMR as well as solvent, TLC solvent and more.

(4)   Please comment the spectral characteristics for the obtained connection.

(5)   Add “microwave irradiation” to keywords.

(6)   Figure 3 – separate of spectrals.

(7)   Avoid lumping references as in [1,2], [6-7] and all other. Instead summarize the main contribution of each referenced paper in a separate sentence.

(8)   Need more latest references. Also the style of reference should be improve (see template).

(9)   The conclusion need to be re-written, are too general. The flow of this chapter is poor. 

The sentence of “The product MS is extensively used in the food and pharmacy industries because of its antiinflammatory activity” is not conclusion for this research.

(10)   The English correction is necessary.

Round 2

Reviewer 1 Report

I appreciate the authors taking my suggestions into account.  I believe that the manuscript is now suitable for publication.

Reviewer 2 Report

The Authors have satisfactory answered all my comments. The manuscript has been improved, reads well, it is well organized and systematic. In my humble opinion, this manuscript should be published in this form in Molecules.